# Human Tissue Analysis of Left Atrial Adipose Tissue and Atrial Fibrillation after Cox Maze Procedure

**DOI:** 10.3390/jcm11030826

**Published:** 2022-02-04

**Authors:** Jung-Hwan Kim, Joon-Young Song, Hyo-Sup Shim, Sak Lee, Young-Nam Youn, Hyun-Chel Joo, Kyung-Jong Yoo, Seung-Hyun Lee

**Affiliations:** 1Division of Cardiovascular Surgery, Department of Thoracic and Cardiovascular Surgery, Severance Cardiovascular Hospital, Yonsei University College of Medicine, Yonsei University Health System, Seoul 03722, Korea; jhkim0907@yuhs.ac (J.-H.K.); songjy2k@yuhs.ac (J.-Y.S.); sak911@yuhs.ac (S.L.); ynyoun@yuhs.ac (Y.-N.Y.); vietcomm@yuhs.ac (H.-C.J.); kjy@yuhs.ac (K.-J.Y.); 2Department of Pathology, Severance Hospital, Yonsei University College of Medicine, Seoul 03722, Korea; shimhs@yuhs.ac

**Keywords:** atrial fibrillation, Cox maze procedure, atrial adipose tissue

## Abstract

Cardiac adipose tissue is a well-known risk factor for the recurrence of atrial fibrillation (AF) after radiofrequency catheter ablation, but its correlation with maze surgery remains unknown. The aim of this study was to investigate the correlation between the recurrence of AF and the adipose component of the left atrium (LA) in patients who underwent a modified Cox maze (CM) III procedure. We reviewed the pathology data of resected LA tissues from 115 patients, including the adipose tissue from CM-III procedures. The mean follow-up duration was 30.05 ± 23.96 months. The mean adipose tissue component in the AF recurrence group was 16.17% ± 14.32%, while in the non-recurrence group, it was 9.48% ± 10.79% (*p* = 0.021), and the cut-off value for the adipose component for AF recurrence was 10% (*p* = 0.010). The rates of freedom from AF recurrence at 1, 3, and 5 years were 84.8%, 68.8%, and 38.6%, respectively, in the high-adipose group (≥10%), and 96.3%, 89.7%, and 80.3%, respectively, in the low-adipose group (<10%; *p* = 0.002). A high adipose component (≥10%) in the LA is a significant risk factor for AF recurrence after CM-III procedures. Thus, it may be necessary to attempt to reduce the perioperative adipose portion of the cardiac tissue using a statin in a randomized study.

## 1. Introduction

Atrial fibrillation (AF) is the most frequent heart rhythm disorder and is known to be associated with obesity, metabolic syndrome, and inflammation [1,2]. Patients with heart valve disease often have AF, of which mitral valve (MV) disease is the most common type, occurring in about 40–60% of heart valve disease cases [3,4]. MV surgery with concomitant AF ablation is superior to MV surgery alone with an intensive rhythm control strategy, and the potential benefits and safety of a surgical ablation procedure for AF during MV surgeries have been well documented [4]. The benefits of one surgical ablation procedure called the Cox maze procedure include the restoration of synchronous atrioventricular contractions, leading to an improvement in cardiac output, the relief of palpitation symptoms, the prevention of thromboembolic events, and the discontinuation of anticoagulation therapy, which may improve the quality of life [3].

However, about 30–35% of patients who undergo maze procedures experience AF recurrence during follow-up [5]; therefore, many studies have attempted to reveal factors that can predict risk or failure, such as AF duration, left atrial size, and the degree of left atrial tissue fibrosis. Epicardial adipose tissue was recently shown to be associated with the prevalence of AF and is considered a risk factor in the maintenance of SR after radiofrequency catheter ablation (RFCA) [6]. Epicardial adipose tissue, as a specialized type of visceral adipose tissue, has also been considered to be an endocrine organ that produces various proinflammatory factors, such as leptin, interleukin 6, adipocytokines, and tumor necrosis factor α [7], and has been suggested to play a significant role in the promotion of arrhythmia due to its proinflammatory properties and anatomical proximity to the myocardium [8].

Based on this correlation between RFCA and AF recurrence, we hypothesized that myocardial adipose tissue would be strongly correlated with the clinical results of the Cox maze III procedure in terms of rhythm outcomes. Therefore, we evaluated the clinical outcomes of patients who underwent a modified Cox maze III procedure by analyzing the association between the adipose tissue component and the degree of fibrosis in the resected left atrium (LA) with AF recurrence.

## 2. Material and Methods

### 2.1. Patient Population and Data Collection

This study initially included 373 patients who underwent a modified Cox maze III procedure between March 2009 and June 2018. Of these patients, 160 underwent a modified Cox maze III procedure with LA resection, and the resected LA tissue samples were subjected to pathological analysis. In total, 17 of the 160 patients’ electrocardiograms (ECGs) were excluded from the study because of irregular follow-ups or a lack of clarity. A further 28 patients failed to maintain SR within 1 to 3 days after surgery and were excluded; we considered these patients unsuitable for analyzing the effect of adipose tissue on AF recurrence. Finally, 115 resected LA samples underwent pathological analysis focusing on the adipose tissue and fibrosis. Data were collected from electronic medical records and analyzed retrospectively. This study was approved by our institutional review board, which waived the requirement for informed consent due to the study’s retrospective nature. Neither patients nor the public were involved in the design, conduct, reporting, or dissemination plans of our research.

### 2.2. Surgical Procedure

Most patients underwent a simplified surgical ablation (based on Cox maze III) with cryoenergy sources (NO gas base, −70~80 °C, CryoICE Cryoablation Probes, Atricure). Modifications to the original Cox maze III procedure consisted of both left- and right-side ablation. Left-side ablations involved pulmonary vein (PV) isolation, the mitral isthmus, left atrial auricle (LAA) isolation, and connections from PV isolation to the LAA. Right-side ablations involved a carvo-carvo lesion (from the superior vena cava to the inferior vena cava), a carvo (intercaval ablation line)-tricuspid annulus isthmus lesion, and the RA free wall. Resection of the LA inferior wall was performed in all cases to reduce the LA volume, and these LA posterior wall tissues were used for pathological analysis in this study.

### 2.3. Pathological Analysis of Adipose Tissue

The surgical specimens were fixed in 10% neutral-buffered formalin and embedded in paraffin blocks. The entire tissue sample was submitted for microscopic examination. From each formalin-fixed, paraffin-embedded tissue block, 4 μm sections were cut and stained with hematoxylin and eosin. The myocardial tissue, adipose tissue, and fibrous tissue were subjected to histological analysis. The percentage of the adipose component was recorded in 5% increments (i.e., 5%, 10%, 15%; Figure 1).

### 2.4. Postoperative Rhythm Follow-Up

After the procedure, the cardiac rhythm was monitored continuously in the intensive care unit. We checked 12-channel surface ECGs every day during hospitalization. An atrial lead was routinely placed on the RA at the end of surgery to allow electroatriography in all patients during atrial lead insertion. After patients were discharged, we checked their ECGs at 3- to 6-month intervals in the outpatient clinic. The frequency of ECG monitoring was thereafter determined by the attending surgeons and cardiologists. The patients who had AF underwent 12-channel ECG monitoring every 3 months in the outpatient clinic, and those with SR underwent the same monitoring once a year. Holter monitoring was usually performed 6 to 12 months after surgery.

Many institutions routinely use 200 mg of amiodarone twice a day for 5 days after surgery, followed by 200 mg daily for 3 months to reduce the risk of early atrial arrhythmia recurrence. However, we did not use amiodarone, but unless contraindicated, a small dose of a beta blocker can be used for a postoperative atrial tachyarrhythmia event if the patient’s cardiac index is tolerable immediately after surgery.

If the rhythm had not converted to SR within 3 months after surgery, the maze surgery was considered to be a failure. AF recurrence was defined at each time point as any documented episode of AF, atrial flutter, or atrial tachycardia (>30 s without use of class I/III antiarrhythmic drugs (AADs) after 3 months of the blanking period from surgery) since the previous electrocardiographic checkpoint. The patients’ lipid profiles were obtained before the Cox maze III procedure to analyze the correlation between the lipid profile and adipose tissue component.

### 2.5. Data Analysis

Continuous variables are presented as the mean ± standard deviation, and categorical variables are presented as percentages or numbers. For univariate analyses, preoperative and operative variables (including the pathological results and the portion of adipose and fibrosis tissues in the resected LA) were analyzed using either the Kaplan-Meier method or a Cox regression model to investigate the influence of the adipose tissue component on the maze failures during the follow-up period. Independent predictors were determined using Cox multivariate analysis. Receiver operating characteristic (ROC) curves were used to obtain the adipose tissue component cut-off value for AF recurrence. The rate of freedom from AF according to the adipose component (high vs. low) was estimated using the Kaplan-Meier method. All reported *p* values of less than 0.05 were considered to indicate statistical significance. IBM SPSS 25 (IBM Corp., Armonk, NY, USA) was used for the statistical analyses.

## 3. Results

### 3.1. Perioperative Characteristics

The mean duration of follow-up was 30.05 ± 23.96 months (interquartile range (IQR): 1–119 months), and the mean patient age was 60.49 ± 10.49 years (IQR: 33 to 81 years). A total of 21 patients (13.1%) had paroxysmal AF, 70 (43.8%) had persistent AF, and 69 (43.1%) had long-standing AF. The mean duration of AF before surgery was 41.02 ± 67.01 months. The patients’ baseline characteristics are summarized in Table 1.

### 3.2. Operative Outcomes

In total, 150 patients (93.8%) underwent a bilateral atrial maze procedure, and 10 patients (6.3%) underwent left-only atrial surgical ablation with LAA isolation. One hundred and fifteen patients (71.9%) maintained a normal SR. Eleven patients (6.9%) underwent an additional postoperative radiofrequency ablation, two of whom underwent radiofrequency ablation within 6 months after surgery. Permanent pacemakers were inserted in nine patients (5.6%). Table 2 presents the detailed data.

### 3.3. Effects of Late Rhythm Outcomes and Adipose Component on AF Recurrence

AF recurred in 32 of the 160 patients (27.83%) during follow-up. The mean time to recurrence was 27.43 ± 27.64 months (IQR: 6.69–112.69 months). A significant difference was found in adipose tissue between patients with recurrence and those without. Furthermore, patients who had AF recurrence were older and had a longer preoperative duration of AF, but the difference was not statistically significant (Table 3). The lipid profiles (cholesterol, low-density lipoprotein, high-density lipoprotein, and triglycerides) showed no significant difference between patients with recurrence and those without. The correlation between the adipose tissue and lipid profile was tested, and none of the lipid profile data showed a significant correlation with the adipose tissue component. Table 3 and Figure 2 present the detailed data. Cox hazard risk analysis revealed that age and a high adipose component were significant risk factors for the recurrence of atrial arrhythmia >30 s without the use of class I/III antiarrhythmic drugs (AADs) after 3 months from surgery (Table 4).

The ROC curve analysis for the effect of adipose tissue on AF recurrence showed that the statistically meaningful cut-off value of the adipose tissue component was 10% (AUC (area under the curve): 0.655; 95% confidence interval (CI): 0.541–0.768; *p* = 0.010; Figure 3); therefore, we divided the samples into groups with high (≥10%) vs. low (<10%) levels of adipose tissue and compared the late rhythm outcomes between the two groups. Kaplan-Meier analysis revealed cumulative AF-free survival rates of 84.8%, 57.5%, and 38.6% in the high-adipose group (≥10%), and of 96.3%, 84.1%, and 80.3% in the low-adipose group (<10% *p* = 0.002), at 1, 3, and 5 years after surgery, respectively (Figure 4).

## 4. Discussion

Although the usefulness of the maze procedure has already been demonstrated, AF tends to recur over time after a successful maze procedure [3]. The outcomes of the maze procedure are influenced by the thoroughness of follow-ups and by the method of rhythm assessment. Considering the method of rhythm assessment, the “last follow-up rhythm” may underestimate the recurrence rate of AF, which then overestimates the success rate of the procedure. Conversely, actuarial methods used to delineate time-related events, such as the “AF recurrence-free rate”, define any recurrent AF as a failure of the procedure, which may underestimate the actual clinical success rate. Because the most favorable method of reporting the success rate is “rhythm at last follow-up”, it was adopted in order to evaluate the success of maze procedures in this study [9,10,11], and the success of the maze procedure was exclusively defined as the maintenance of SR after the maze procedure. In contrast, patients with recurring atrial tachycardia and a cardiac rhythm controlled with a permanent pacemaker were categorized into the recurrent AF group.

AF recurrence exposes patients to additional discomfort and cardiovascular events and leads to hospitalization and other interventions that impose a relevant economic burden on health care systems [3,12]. The well-known risk factors for AF recurrence after a maze procedure include old age, a larger left atrial diameter, a longer history of AF, a lower-amplitude f-wave, rheumatic MV disease, long-standing AF, and lesion sets of maze procedures [1,13]. In this study, the portion of the adipose component in the LA was identified as a strong independent predictor of AF recurrence after a Cox maze III procedure. The survival rates of the high- and low-adipose groups revealed statistically significant differences. To the best of our knowledge, no association between AF recurrence and the adipose component of LA tissue in the modified Cox maze III procedure has yet been clearly identified.

A recent study reported the underlying mechanisms by which epicardial adipose tissue (EpAT) influences the atrial substrate for AF [14]. The study showed that the local EpAT content was associated with conduction slowing and increased the complexity of activation patterns, accompanied by increased cardiac fibrosis. Furthermore, the study showed adipose tissue infiltration was associated with conduction heterogeneity. These changes are associated with the release of a complement of proteins with the capacity to disrupt intermyocyte adhesion, modulate cellular metabolism, and increase inflammation. The authors concluded that these findings indicate that local epicardial adiposity influences the adjacent myocardium and promotes functional heterogeneity, which contribute to conduction abnormalities underlying the AF substrate.

In this study, the meaningful cut-off value for the adipose tissue component was 10%. The adipose tissue data were derived from biopsies. The adipose component portion according to AF type (paroxysmal, persistent, and long-standing) was not different between groups, but the recurrent AF group showed higher levels of the adipose component compared with the SR-maintaining group (Appendix A). Magnetic resonance imaging (MRI) and computed tomography (CT) are considered as the standard modalities for the assessment of total body fat and the quantification of adipose tissue, but the difficulty of standardizing measurement locations limits the determination of fat thickness reference values, and artifacts such as noise, beam, hardening, scatter, pseudo-enhancement, motion, cone beam, helical, ring, and metal artifacts often interfere with measurements [15,16]. In contrast, biopsies have a strong advantage in accurately determining the thickness and content of adipose tissue.

Recent studies have suggested that treatment with statins significantly reduces the incidence of new-onset or recurrent AF in primary and secondary prevention; therefore, caution is recommended in extrapolating these findings [17]. However, no significant correlations were observed between the adipose component and the lipid profiles in our study. Most patients had stable lipid profile values (within the normal range), and statin intake was not analyzed in this study; therefore, a larger study population is needed to more closely estimate the correlation between adipose tissue and the lipid profile.

Our study has several potential limitations. First, this is a retrospective observational data study. Second, we did not determine the exact AF recurrence date or timing because our analysis was a cross-sectional cohort study, and we did not continuously monitor the patients’ ECGs. Therefore, it is possible that patients who were in SR at the outpatient clinic might experience atrial arrhythmia in their daily life and vice versa. Finally, pathological analysis of the LA tissue cannot represent the overall myocardial or epicardial fat tissue burden; therefore, the LA adipose component itself can be questioned in the analysis of AF recurrence. However, LA tissue can easily be taken during the Cox maze III procedure, which is an especially important part of the maze procedure’s success. Therefore, LA tissue is very valuable and important for the study of arrhythmia pathology. Furthermore, some patients needed beta blockers, not for arrhythmia, but for hypertension or other purposes, so beta blocker usage analysis was very difficult, and we need more data collection surrounding beta blocker usage, despite there being few usage cases.

## 5. Conclusions

A high adipose component (≥10%) in the LA was associated with a higher recurrence rate of AF in the patients who underwent the modified Cox maze III procedure. Therefore, we must closely check the adipose tissue burden of cardiac tissue during maze surgeries to modify surgical strategies such as the ablation time. It may thus be necessary to attempt to reduce the perioperative adipose portion of the cardiac tissue using a statin in a randomized study.

## Figures and Tables

**Figure 1 jcm-11-00826-f001:**
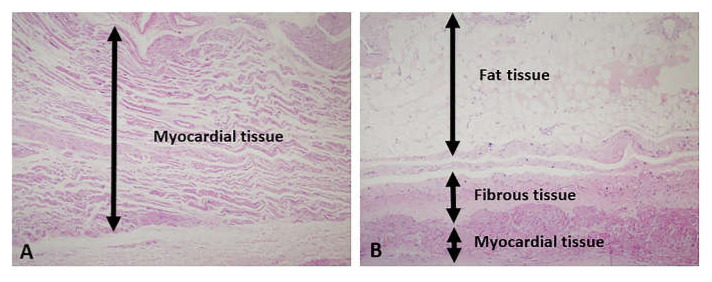
Hematoxylin and eosin staining of the left atrial tissue ((**A**) adipose tissue component <10%; (**B**) adipose tissue component ≥10%).

**Figure 2 jcm-11-00826-f002:**
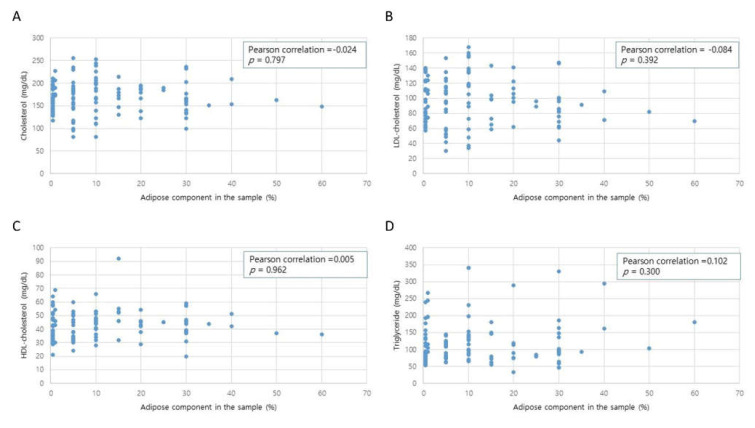
Linear correlation analysis between the adipose component and the laboratory lipid profiles ((**A**) total cholesterol; (**B**) low-density lipoprotein cholesterol; (**C**) high-density lipoprotein cholesterol; (**D**) triglyceride).

**Figure 3 jcm-11-00826-f003:**
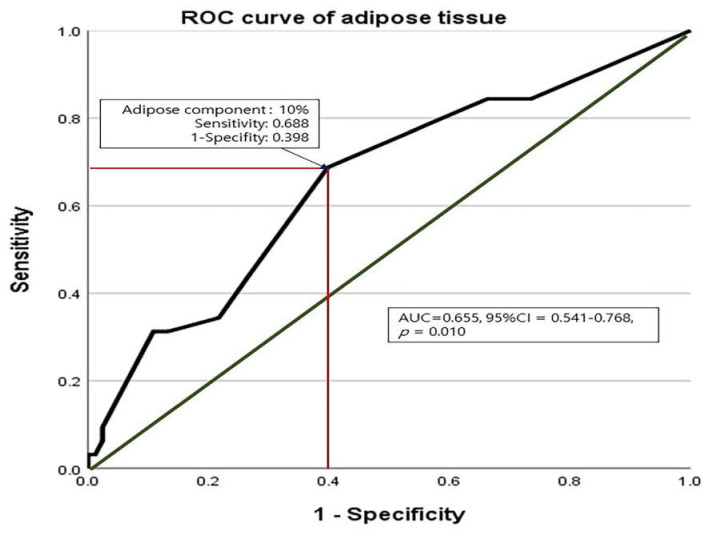
The receiver operating characteristic curve for the effect of left atrial adipose tissue on atrial fibrillation recurrence.

**Figure 4 jcm-11-00826-f004:**
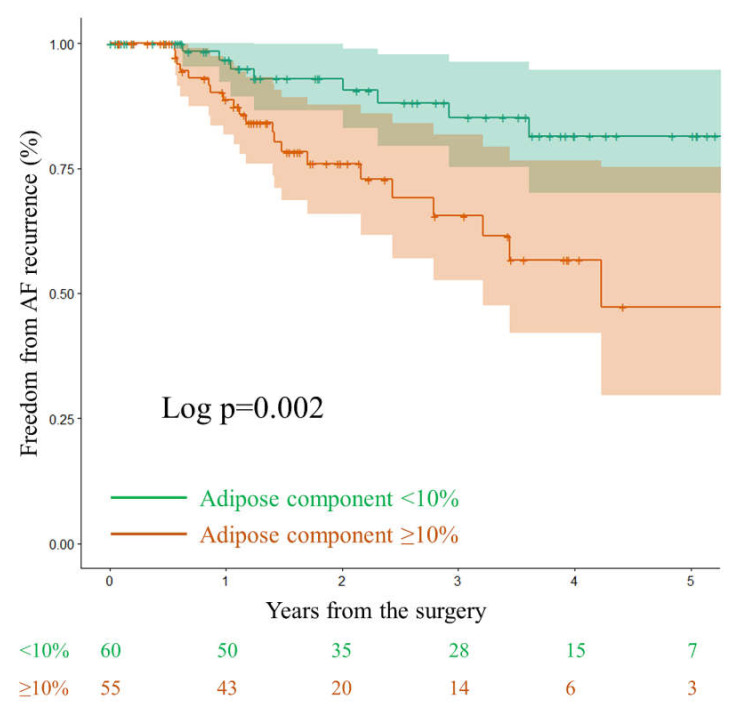
Freedom from atrial fibrillation recurrence rate curves for the left atrial high-adipose component group (≥10%, orange line) and low-adipose component group (<10%, green line) using the Kaplan-Meier method with 95% confidence intervals.

**Table 1 jcm-11-00826-t001:** Patients’ preoperative characteristics.

Variables	N = 115
Age, years	60.49 ± 10.49
Male	56 (35%)
AF duration, months	41.02 ± 67.01
AF type	
Paroxysmal	21 (13.1%)
Persistent	70 (43.8%)
Long-standing	69 (43.1%)
Body mass index, kg/m^2^	23.60 ± 3.26
Smoking	151 (94.4%)
Hypertension	104 (65%)
Diabetes mellitus	33 (20.6%)
Chronic kidney disease	9 (5.6%)
Stroke history	41 (25.6%)
Peripheral vascular disease	3 (1.9%)
Coronary artery disease	21 (13.3%)
Preoperative RF ablation	6 (3.8%)
Cholesterol, mg/dL	169.18 ± 36.280
LDL cholesterol, mg/dL	97.46 ± 31.597
HDL cholesterol, mg/dL	43.41 ± 10.707
Triglyceride, mg/dL	119.11 ± 64.394

AF, atrial fibrillation; RF ablation, radiofrequency ablation; LDL, low-density lipoprotein; HDL, high-density lipoprotein.

**Table 2 jcm-11-00826-t002:** Operative data.

Variables	N = 115
Concomitant surgery	
Aortic valve replacement	48 (30%)
Mitral valve repair or replacement	148 (92.5%)
Tricuspid annuloplasty	122 (76.3%)
Type of maze	
Left atrium only	10 (6.3%)
Left and right atrium	150 (93.8%)
Ejection fraction, %	
Preoperative	59.53 ± 11.38
Postoperative	58.71 ± 8.79
Left atrial volume index, mL/m^2^	
Preoperative	101.04 ± 46.63
Postoperative	73.89 ± 30.83
ICU stay, days	3.38 ± 6.19
Hospitalization period, days	13.92 ± 12.14
RF ablation for AF recurrence	9 (7.8%)
Permanent pacemaker insertion	7 (6.1%)

ICU, intensive care unit; PPM, permanent pacemaker.

**Table 3 jcm-11-00826-t003:** Comparisons between the sinus group and the atrial fibrillation recurrence group.

Variables	No Recurrence (***n*** = 83)	Recurrence (***n*** = 32)	***p*** Value
Age, years	57.59 ± 10.49	61.78 ± 10.19	0.055
Male	27 (32.5%)	9 (28.1%)	0.648
Body mass index, kg/m^2^	23.78 ± 3.09	23.52 ± 3.29	0.686
AF type			0.240
Paroxysmal	14 (16.9%)	3 (9.4%)	
Persistent	41 (49.4%)	13 (40.6%)	
Long-standing	28 (33.7%)	16 (50.0%)	
Smoking	80 (96.4%)	29 (90.6%)	0.346
Hypertension	57 (68.7%)	21 (65.6%)	0.754
Diabetes mellitus	16 (19.3%)	6 (18.8%)	0.949
Chronic kidney disease	2 (2.4%)	2 (6.3%)	0.309
CVA history	17 (20.5%)	8 (25.0%)	0.599
Coronary artery disease	9 (10.8%)	2 (6.3%)	0.725
Total cholesterol, mg/dL	172.80 ± 35.918	159.91 ± 36.102	0.088
LDL cholesterol, mg/dL	99.70 ± 30.851	82.10 ± 33.211	0.263
HDL cholesterol, mg/dL	43.41 ± 11.275	43.42 ± 9.387	0.995
Triglyceride, mg/dL	124.19 ± 72.452	107.00 ± 37.236	0.113
Adipose tissue, %	9.48 ± 10.79	16.17 ± 14.32	0.021
Fibrosis, %	5.08 ± 9.19	5.656 ± 9.81	0.770
Type of maze			0.671
LA only	6 (7.2%)	1 (3.1%)	
Both atriums	77 (92.8%)	31 (96.9%)	
Ejection fraction, %	59.06 ± 11.28	60.84 ± 13.59	0.849
LAVI, mL/m^2^	93.45 ± 33.42	96.80 ± 33.67	0.641

AF, atrial fibrillation; CVA, cerebrovascular accident; LDL, low-density lipoprotein; HDL, high-density lipoprotein; LA, left atrium; LAVI, left atrial volume index.

**Table 4 jcm-11-00826-t004:** Cox regression analysis for atrial fibrillation recurrence.

Variables	Univariate		Multivariate	
	HR (95% CI)	*p* Value	HR (95% CI)	*p* Value
Female	1.436 (0.641–3.217)	0.380		
Age, per year	1.055 (1.015–1.096)	0.007	1.045 (1.005–1.087)	0.026
Persistent vs. paroxysmal	1.219 (0.346–4.297)	0.758		
Long-standing vs. paroxysmal	2.112 (0.605–7.373)	0.241		
Body mass index, per kg/m^2^	1.063 (0.940–1.201)	0.329		
Smoking	4.834 (0.566–41.253)	0.150		
Hypertension	0.850 (0.399–1.814)	0.675		
Diabetes mellitus	0.886 (0.360–2.178)	0.792		
Chronic kidney disease	2.680 (0.628–11.442)	0.183		
CVA history	0.166 (0.730–3.755)	0.227		
Coronary artery disease	0.583 (0.139–2.457)	0.463		
Total cholesterol, per mg/dL	0.994 (0.984–1.004)	0.240		
LDL cholesterol, per mg/dL	0.996 (0.984–1.007)	0.444		
HDL cholesterol, per mg/dL	0.994 (0.963–1.027)	0.726		
Triglyceride, per mg/dL	0.997 (0.990–1.004)	0.358		
Adipose tissue	1.045 (1.021–1.070)	<0.001	1.046 (1.021–1.071)	<0.001
Fibrosis	1.016 (0.984–1.048)	0.336		
AVR	1.487 (0.656–3.373)	0.342		
MVR or MVP	1.195 (0.161–8.847)	0.862		
TAP	0.687 (0.305–1.550)	0.366		
Maze type				
LA only vs. both atriums	1.869 (0.251–13.940)	0.542		
Ejection fraction, per %	1.003 (0.973–1.033)	0.865		
LAVI, per ml/m^2^	1.005 (0.994–1.016)	0.396		

CVA, cerebrovascular accident; LDL, low-density lipoprotein; HDL, high-density lipoprotein; AVR, aortic valve replacement; MVR, mitral valve replacement; MVP, mitral valvuloplasty; TAP, tricuspid annuloplasty; LA, left atrium; LAVI, left atrial volume index.

## Data Availability

The data presented in this study are openly available.

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
