# Peer review of "Human Tissue Analysis of Left Atrial Adipose Tissue and Atrial Fibrillation after Cox Maze Procedure"

_jcm, 2022, doi:10.3390/jcm11030826_

Round 1
Reviewer 1 Report
The paper presents notes of originality. Appropriate materials and methods. Accurate statistical analysis. I recommend certified and accurate revision of the English language
Author Response
Point 1: The paper presents notes of originality. Appropriate materials and methods. Accurate statistical analysis. I recommend certified and accurate revision of the English language
Response 1: Thank you for your comments. Our manuscript was reviwed by an English editor for English proofreading once again.
Reviewer 2 Report
Jung-Hwan Kim et al described the correlation between left atrial adipose and af recurrence after cox maze procedure. The paper is well written and interesting. It has the potential to become a background for future prospective trials with bigger clinical impact. I have only few comments:
- I think that the AF types should be categorized differently. By the definition from 2020 ESC Guidelines permanent AF means: "AF that is accepted by the patient and physician, and no further attempts to restore/maintain sinus rhythm will be undertaken." Taking into consideration, that in patients described in the study actions were made to restore SR I think 'Long-standing' would suit better.
- The conclusions in the abstract and in the main bode are slightly different, please unify.
- I am aware of the difficulties in obtaining data in the retrospective details. However, an effort should be made to present the data regarding post-operative treatment (beta blockers and eventually other anti-arrhythmic drugs). Maybe the effect was also dependent from the concomitant treatment?
- I did not understand: if patient had an AF recurrence at some point after the procedure, but presented with SR at the ambulatory visit - was it categorized as recurrence or not? Please clarify.
- Authors state: 'The lipid profiles showed no significant difference between the patients with recurrence and those without.' and later in the conclusion: 'We should thoroughly analyze laboratory lipid profiles and adipose tissue burden before surgery.'. I don't see justification for lipid profile analysis based on the presented data.
Author Response
Point 1: I think that the AF types should be categorized differently. By the definition from 2020 ESC Guidelines permanent AF means: "AF that is accepted by the patient and physician, and no further attempts to restore/maintain sinus rhythm will be undertaken." Taking into consideration, that in patients described in the study actions were made to restore SR I think 'Long-standing' would suit better.
Response 1: Thank you for your important comment. I agree with your comments and I changed from ‘permanenet AF’ to ‘long-standing AF’ in manuscript.
Point 2: The conclusions in the abstract and in the main bode are slightly different, please unify.
Response 2: Thank you for your insightful comment. We modified the conclusion of manuscript to unify with abstract.
Point 3: I am aware of the difficulties in obtaining data in the retrospective details. However, an effort should be made to present the data regarding post-operative treatment (beta blockers and eventually other anti-arrhythmic drugs). Maybe the effect was also dependent from the concomitant treatment?
Response 3: Thank you for your very important comment. In Method section (line 104), there are some comments about our strategy for medication. It was “Many institutions used routinely amiodarone at 200 mg twice a day for 5 days after surgery followed by 200 mg daily for 3 months to reduce the risk of early atrial arrhythmia recurrence. However, we don`t use amiodarone as possible, but unless contraindicated, small dose of beta blocker can be used for postoperative atrial tarchy-arrhythmia event if patient`s cardiac index is tolerable immediate after surgery.”
There were 6 patients who received anti-arrhythmic drug (amiodarone) and 8 patients who received beta blocker, which was too small number of patients to affect to the results.
Point 4: I did not understand: if patient had an AF recurrence at some point after the procedure, but presented with SR at the ambulatory visit - was it categorized as recurrence or not? Please clarify.
Response 4: AF recurrence was defined as any documented episode of AF, atrial flutter, or atrial tachycardia after 3-months blanking period from the surgery. If the any epiosed of AF, atrial flutter, or atrial tachycardia were documented after 3-months blanking period from the surgery, we defined as ‘recurrence’. But if those episode which was not documented, we defined as sinus rhythm.
Point 5: Authors state: 'The lipid profiles showed no significant difference between the patients with recurrence and those without.' and later in the conclusion: 'We should thoroughly analyze laboratory lipid profiles and adipose tissue burden before surgery.'. I don't see justification for lipid profile analysis based on the presented data.
Response 5: Thank you for your insightful comment. I agree with your comments. In our data, it is difficult to conclude that lipid profile can affect to the recurrence. So we removed that comments from conclusion.

Reviewer 3 Report
Kim et al. highlight in the current manuscript the association of cardiac adipose tissue, by classifying adipose content on resected LA tissues, and recurrent AF freedom in patients undergoing the Cox Maze procedure. Their findings demonstrate that increased age and adipose tissue were independent risk factors for AF recurrence. Notably, higher (>10%) adipose tissue component in the atrial biopsies was correlated to increased rate of recurrent AF following the surgical ablation over 5 years post-surgery. Atrial adipose tissue burden which can be assessed in resected LA biopsy (commonly performed during surgical procedure), may be useful in predicting patients' freedom from AF post-surgery and guide therapies in SR maintenance
Comments:
1) Epicardial adipose tissue (EAT) is a recently identified risk factor for AF recurrence following radiofrequency catheter ablation strategies (Ref 6), itself associated with obesity (Aitken-Buck et al., Adipocyte, 2019; PMID: 31829077), of which obesity is a well-known for AF development. It is therefore striking that the patient cohort had normal BMI (~23) in both AF recurrent and SR maintained populations despite the differences in EAT % (<10% vs >10% as a cut-off). Can the authors please comment further on this?
2) The authors have not discussed mechanisms behind how EAT can lead to AF. I refer the authors to Nalliah et al., JACC, 2020 (PMID: 32883413) which describes conduction abnormalities associated with EAT.
3) Line 151 - "A significant difference was found in age, adipose tissue and preoperative duration of AF between the patients with recurrence and those without." I could not find details on the differences preoperative duration of AF between the two cohorts in Table 3, and Age, although trended towards significance, was p=0.055 (Table 3) which the authors determined statistical significance to be p<0.05.
4) Line 213 - "..., but recurred group regardless of AF type showed higher adipose component..." I note that in Supplement Table 1, only Paroxysmal and Persistent showed significantly higher adipose component in the recurred group compared to SR whereas it was not significant in the Permanent AF group (P=0.50).
Author Response
Point 1: Epicardial adipose tissue (EAT) is a recently identified risk factor for AF recurrence following radiofrequency catheter ablation strategies (Ref 6), itself associated with obesity (Aitken-Buck et al., Adipocyte, 2019; PMID: 31829077), of which obesity is a well-known for AF development. It is therefore striking that the patient cohort had normal BMI (~23) in both AF recurrent and SR maintained populations despite the differences in EAT % (<10% vs >10% as a cut-off). Can the authors please comment further on this?
Response 1: Thank you for your important comment. As your comments, it is well known that obesity is one of the strong risk factor for AF progression. But, I suppose that our study has too small number of patients to make a statistical difference. And our included patients was highly selected patients, so there is a possibility of selection bias.
Point 2: The authors have not discussed mechanisms behind how EAT can lead to AF. I refer the authors to Nalliah et al., JACC, 2020 (PMID: 32883413) which describes conduction abnormalities associated with EAT.
Response 2: Thank you for your very important comment. I agree with your comment that mechanisms how adipose tissue can lead to AF. So we reviewed above article and added in conclusion saction as follwing:
‘Recent study reported the underlying mechanisms by which epicardial adipose tis-sue (EpAT) influences the atrial substrate for AF [14]. The study showed that local EpAT content associated with conduction slowing and increased complexity of activation pat-terns, accompanied by increased cardiac fibrosis. Furthermore, adipose tissue infiltration associated with conduction heterogeneity. These changes associated with the release of a complement of proteins with capacity to disrupt intermyocyte adhesion, modulate cellu-lar metabolism, and increase inflammation. The authors concluded that these findings indicated that local epicardial adiposity influences adjacent myocardium and promotes functional heterogeneity, which contributes to conduction abnormalities underlying the AF substrate.’
Point 3: Line 151 - "A significant difference was found in age, adipose tissue and preoperative duration of AF between the patients with recurrence and those without." I could not find details on the differences preoperative duration of AF between the two cohorts in Table 3, and Age, although trended towards significance, was p=0.055 (Table 3) which the authors determined statistical significance to be p<0.05.
Response 3: Thank you for your important comment. There were some mistakes describing the data. So, we changed to ‘A significant difference was found in adipose tissue between the patients with recurrence and those without. And patients who had AF recurrence were older and had longer preoperative duration of AF, but the difference was not statistically significant (table 3).’
Point 4: Line 213 - "..., but recurred group regardless of AF type showed higher adipose component..." I note that in Supplement Table 1, only Paroxysmal and Persistent showed significantly higher adipose component in the recurred group compared to SR whereas it was not significant in the Permanent AF group (P=0.50).
Response 4: Thank you for your important comment. The p-value of permanent AF was 0.049 and 0.05 after round off. The 0.05 was mistake. We changed the supplement table 1.
